# Citrullinated ENO1 Vaccine Enhances PD-1 Blockade in Mice Implanted with Murine Triple-Negative Breast Cancer Cells

**DOI:** 10.3390/vaccines13060629

**Published:** 2025-06-11

**Authors:** Ricardo A. León-Letelier, Alejandro M. Sevillano-Mantas, Yihui Chen, Soyoung Park, Jody Vykoukal, Johannes F. Fahrmann, Edwin J. Ostrin, Candace Garrett, Rongzhang Dou, Yining Cai, Fu-Chung Hsiao, Jennifer B. Dennison, Eduardo Vilar, Banu K. Arun, Samir Hanash, Hiroyuki Katayama

**Affiliations:** 1Department of Clinical Cancer Prevention, The University of Texas MD Anderson Cancer Center, Houston, TX 77030, USA; raleon@mdanderson.org (R.A.L.-L.); a.sevillano@icf.com (A.M.S.-M.); ychen43@mdanderson.org (Y.C.); snpark@mdanderson.org (S.P.); jvykouka@mdanderson.org (J.V.); jffahrmann@mdanderson.org (J.F.F.); csgarrett@mdanderson.org (C.G.); rdou@mdanderson.org (R.D.); ycai4@mdanderson.org (Y.C.); fchsiao@mdanderson.org (F.-C.H.); jbdennis@mdanderson.org (J.B.D.); evilar@mdanderson.org (E.V.); shanash@mdanderson.org (S.H.); 2Therapeutic Innovation Center, Baylor College of Medicine, Houston, TX 77030, USA; 3Department of General Internal Medicine, The University of Texas MD Anderson Cancer Center, Houston, TX 77030, USA; ejostrin@mdanderson.org; 4Department of Breast Medical Oncology, The University of Texas MD Anderson Cancer Center, Houston, TX 77030, USA; barun@mdanderson.org

**Keywords:** cancer vaccine, post-translational modification, immunopeptidome, triple-negative breast cancer, PD-1 blockade

## Abstract

**Background/Objectives**:Cancer vaccine targets mostly include mutations and overexpressed proteins. However, cancer-associated post-translational modifications (PTMs) may also induce immune responses. Previously, our group established the enzyme protein arginine deiminase type-2 (PADI2), which catalyzes citrullination modification, is highly expressed in triple-negative breast cancer (TNBC), promoting antigenicity. **Methods**: Here, we show the workflow of designing citrullinated enolase 1 (citENO1) vaccine peptides identified from breast cancer cells by mass spectrometry and demonstrate TNBC vaccine efficacy in the mouse model. Immunized mice with citENO1 peptides or the corresponding unmodified peptides, plus Poly I:C as an adjuvant, were orthotopically implanted with a TNBC murine cell line. **Results**: Vaccination with citENO1, but not unmodified ENO1 (umENO1), induced a greater percentage of activated CD8+ PD-1+ T cells and effector memory T cells in skin-draining lymph nodes (SDLNs). Remarkably, the citENO1 vaccine delayed tumor growth and prolonged overall survival, which was further enhanced by PD-1 blockade. **Conclusions**: Our data suggest that cancer-restricted post-translational modifications provide a source of vaccines that induce an anti-cancer immune response.

## 1. Introduction

Triple-negative breast cancer (TNBC) is a subtype of breast cancer that has a high metastasis rate and poor prognosis [1,2]. Chemotherapy remains the primary choice of systemic treatment for TNBC [3,4]. However, response to chemotherapy is variable and lacks durability, and acquired resistance is a frequent manifestation. Immunotherapy is an emerging alternative therapeutic strategy for TNBC, like the use of the PD-1 blockade, which has been recently approved for patients in the neoadjuvant setting [5]. Yet, only a subset of patients benefit from immune checkpoint inhibitor (ICI) therapy, necessitating alternative strategies. Cancer vaccines that elicit strong reactivity to well-defined tumor antigens hold potential as a promising preventive strategy or a means for combination therapy [6,7]. Considering that BRCA1 mutations contribute to TNBC development, personalized preventative cancer vaccines for individuals with germline BRCA1 mutation could be beneficial. Targets for cancer vaccines include mutated proteins and peptides as well as proteins that are overexpressed in cancer but are otherwise ubiquitous in their structure, as well as cryptic peptides [8]. More recently, there have been several mRNA-based cancer vaccine approaches in clinical trials for different cancer types [9,10]. However, protein alterations other than mutations, notably post-translational modifications (PTMs), may induce an immune response. Moreover, mRNA-based vaccines do not address PTMs. Several lines of evidence support the role of citrulline protein modifications in inducing immunogenicity [11,12,13]. Dysregulated protein citrullination by peptidyl arginine deiminases (PADIs) is currently being explored for its relevance in cancer [13]. PADIs comprise a family of enzymes that, in the presence of calcium ions, catalyze the post-translational modification of proteins via the deamination of arginine to citrulline. In total, five PADI family members are known, with sequence homology ranging from 70% to 95% [13]. To date, no enzyme that can reverse citrullination has been identified. The role of protein citrullination has been best investigated in the context of rheumatoid arthritis (RA), wherein elevated protein citrullination has been observed in the synovial fluid [14]. Autoimmunity in RA is considered to be principally facilitated through the major histocompatibility complex (MHC) class II-mediated presentation of citrullinated peptides that elicit a CD4+ T and B cell response. Currently, there is increased interest in MHC-II neoantigens shaping tumor immunity [15]. Moreover, protein citrullination in cancer cells can occur as a result of cellular stress-induced autophagy [16]. Unlike in autoimmune diseases, citrullination in cancer is essentially limited to tumor cells. Autophagosomes, developed during autophagy, provide citrullinated proteins for antigen presentation and act as key drivers for loading antigens onto MHC molecules [17,18].

In a previous study, we established that peptidyl arginine deiminase 2 (PADI2) is highly expressed in TNBC compared to normal mammary and other tissues [19]. We further reported that PADI2-associated protein citrullination promoted antigenicity [19], supporting the role of PADI enzymes in promoting an immune response in cancer [11,12,13]. Surface-expressed ENO1 acts as a plasminogen receptor implicated in tumor cell invasion and metastasis [20]. Silencing ENO1 expression or blocking the interaction between ENO1 and plasminogen resulted in the total inhibition of plasminogen-dependent cell migration in vitro in pancreatic cell lines and the inhibition of metastasis in vivo in a PDAC mouse model [20,21], demonstrating that ENO1 plays an essential role in tumor progression and spread. We found that citrullinated ENO1 (citENO1) was highly enriched on the surface of TNBC cells [19]. Our findings provide a rationale for testing the antigenicity of citENO1 peptides as a cancer vaccine, given the additional supporting evidence in the literature [11,22,23].

## 2. Materials and Methods

### 2.1. Proteomic Analysis

A total of 26 breast cancer cell lines, i.e., MDA-MB-231, HCC1143, HCC1937, HCC1599, HCC1806, MDA-MB-468, HCC70, HCC1187, Hs578T, BT549, HCC1395, HCC38, MDA-MB-436, BT20, MDA-MD-157, SKBR3, HCC1954, HCC202, HCC2218, MCF7, HCC1500, T47D, ZR75-1, CAMA1, BT474, and MDA-MB-361, were obtained from ATCC (Manassas, VA, USA). HLA typing on the respective cell lines was performed using Cellosaurs (www.cellosaurus.org/) (Appendix A), whereas the gene expression data of HLAs were obtained from CCLE (sites.broadinstitute.org/ccle/). Trypsin digestion was applied to surface proteins enriched using EZ-Link™ Micro Sulfo-NHS-SS-Biotinylation Kit (ThermScientific, Waltham, MA, USA) and total cell extracts (TCEs), followed by nano-liquid chromatography (nanoLC)–mass spectrometry, as previously described [24,25,26,27,28,29]. The absolute quantitative (femto-mol) level of ENO1 was estimated from the normalized spectral abundance factor (NSAF) method with a 500 ng tryptic peptide injection [26]. A total of 5 × 10^8^ HCC1954 and TNBC MDA-MB-468 cells were cultured for citrullinated peptide analysis following mild acid elution (MAE) enrichment using EASY-nanoLC-Orbitrap ELITE (ThermoScientific, MA, USA) mass spectrometry, as previously described [30]. The cell line surface, TCE, and mild-acid-eluted peptide data were searched against Sequest HT 1.4 using the Uniprot Human database. One fixed modification of propionamide at Cys (71.037114 Da) and variable modifications, oxidation at Met (15.9949 Da) and deamidation at Arg (0.984016 Da), were added in the search and filtered with an FDR = 0.01. The peptides with deamidated Arg at the C-terminal of the tryptic peptides on the surface and TCE compartments were considered as false identification and removed. Peptide with amino acid lengths greater than 11 were considered as MHC II peptides, and with lengths of 8 to 10 were considered as MHC I peptides [19]. The plasma-immunoglobulin-bound protein analysis of triple-negative breast cancer (TNBC) patients was performed using nano-ACQUITY-Synapt G2-si mass spectrometry (WATERS, Milford, MA, USA), The acquired LC-HDMSE data were processed and searched against the Uniprot Human database through the ProteinLynx Global Server (WATERS, MA, USA) with a false discovery rate of 4% [19,24]. The modification search settings and the deamidated Arg, as previously described with the C-terminal miss cleavage assessment, were the same as in the cell line search.

We performed in silico MHC-II binding affinity predictions using the prediction tool NetMHCIIpan V.3.2. [31,32] and NetMHCpan V4.0. [33]. For these analyses, citrullinated peptide sequences that were converted to the unmodified form were loaded to the respective prediction tool, and the data were sorted based on the binding peptide core affinity prediction (IC50, nM) and % rank with MHC-II and MHC-I assessed with the artificial intelligent network SSNAlignment, and the percentile rank that was generated by comparing the peptide’s score against the scores of one million random 15-mers selected from the Swiss-Prot database. The binding strength threshold based on % rank was as follows: 1% (strong) and 5% (weak) for MHC-II, and 0.5% (strong) and 2% (weak) for MHC-I. The IC50 affinity criteria was 150 nM and 500 nM in MHC-I and MHC-II, respectively. The vaccine ENO1 peptides, ranging from 22 to 24 amino acid length were designed from the citrullination sites identified in the cell surface compartment, mild acid elution peptides and TNBC plasma-immunoglobulin-bound forms, considering the affinity prediction of MHC-II shown in Appendix A.

### 2.2. Gene Expression Datasets for Breast Tumors and Normal Mammary Tissues

Gene expression data and associated clinical information for 1084 breast cancer tumors and 112 normal mammary tissues were obtained from TCGA and the Gene Expression Profiling Interactive Analysis (GEPIA2) web server [34].

### 2.3. Triple-Negative Breast Cancer Cell Model

We used a cell line obtained from a breast tumor model with the conditional knockout expression of the *BRCA1* gene and the haplo-insufficient expression of p53, known as *BRCA1*^CO/CO^; MMTV-Cre; p53^+/−^ (hereafter referred to as BRCA/TP53 for brevity). We also used BRCA/TP53 TNBC cells, which were established to constitutively express luciferase (BRCA/TP53-Luc) [35]. The BRCA/TP53-Luc cell line was cultured in RPMI containing 10% FBS and 0.1% penicillin/streptomycin. Breast tumors were established by the orthotopic injection of 1.5 × 10^6^ cells in the left fat mammary pad of the same genetic background mice that BRCA/TP53 cell line came from, which is B6129SF1/J [35]. Tumors width and length were measured using a caliper every two days starting on day 10. Tumor volume was calculated as (width^2^ × length)/2 in mm^3^. Mice were euthanized if they exhibited cachexia or if the tumor had a length of 2 cm or greater.

### 2.4. Immunohistochemistry (IHC)

Tumors from BRCA/TP53-bearing B6129SF/J mice and the mammary fat pad from naïve B6129SF/J mice were collected for IHC at the Research Histology Core Laboratory, MD Anderson Cancer Center. Tissue slides were deparaffinized in xylene, rehydrated in a descending ethanol series, and then treated with 3% hydrogen peroxide for 10 min. Antigen retrieval was conducted in a pressure cooker in 1X ImmunoRetriever with citrate pH = 6 (Bio SB, Santa Barbara, CA, USA) and 0.1% Tween 20 at 121 °C for 15 min, followed by blocking with 5% goat serum in TBST for 1 h at room temperature. The Ready Probes^TM^ Mouse IgG Blocking Solution (Invitrogen, Waltham, MA, USA, R37621) was used according to the manufacturer’s protocol. To evaluate PADI2, citENO1, ENO1, and PD-L1 expression in tissue sections, anti-PADI2 (Proteintech, Rosemon, IL, USA, Cat#66386-1-Ig, RRID:AB_2881762), anti-citENO1 (Cayman Chemical, Ann Arbor, MI, USA, Cat# 34123, RRID:AB_3532161), anti-ENO1 (Sigma-Aldrich, St. Louis, MO, USA, Cat#WH0002023M1, RRID:AB_1841471), and anti-PD-L1 (R&D Systems, Minneapolis, MN, USA, Cat#AF1019, RRID:AB_354540) mouse antibodies were used at 1:400 dilution overnight at 4 °C. After washing with TBST for 5 min × 3, the secondary antibody was added (Simple Stain MAX PO (MULTI) Universal Immuno-peroxidase Polymer Anti-Mouse and -Rabbit, N-Histofine, Nichirei Bioscience Inc., Chuo-ku, Tokyo, Japan) for 1 h at room temperature. Signal development was performed using an N-Histofine DAV-2V kit (N-Histofine, Nichirei Bioscience Inc., Chuo-ku, Tokyo, Japan).

IHC slides were scanned by the Research Histology Core Laboratory and analyzed using the Aperio Imagescope v12.4.6.5003 (Leica Biosystems, Buffalo Grove, IL, USA). For IHC evaluation, we used the algorithm Positive Pixel Count 11 August 2004 to determine strong (red), medium (orange), weak (yellow), and negative (blue) signals (hue value = 0.1; hue width = 0.5; color saturation threshold = 0.04; lwp (high) = 220; lwp (low) = lp (high) 175; lp (low) = lsp (high) 100; lsp (low) = 0; and lnp (high) = −1). The H-score was subsequently calculated as follows: (3 × % strong signal) + (2 × % medium signal) + (1 × % weak signal), with a score ranging from 0 to 300 from the whole slide or from six different regions for a given sample. The H-scores were subsequently used for statistical analyses.

### 2.5. Immunization with ENO1 Peptides Followed by Anti-PD1 Therapy

The animal experimental protocol was approved by the University of Texas MD Anderson Cancer Center IRB, in accordance with the Guidelines for the Care and Use of Laboratory Animals published by the NIH (Bethesda, Rockville, MD, USA). B6129SF/J mice were obtained from Jackson Laboratory (Cat. #0101043) and housed in a modified barrier. All mice were female and age-matched (8–10 weeks).

We immunized mice with a mix of three citrullinated ENO1 peptides or their unmodified counterpart (CPC Scientific, Rocklin, CA, USA), 10 nmol of each peptide, as the antigen, and Poly (I:C) HMW (InvivoGen, San Diego, CA, USA, tlrl-pic), as the adjuvant (10 µg), in a PBS solution, which was subcutaneously (s.c.) administered in the left flank. The mass spectral and HPLC analyses of the peptides were found in their Certificate of Analysis (Appendix A). Injections were performed weekly and three times in total, based on the dosage used in a previous citENO1 vaccine study [11]. As an additional control, we used PBS alone. Mice were challenged with BRCA/TP53 seven days after the last round of immunization.

For the combination of cancer vaccine and immunotherapy, tumor implantation was performed one week after the last round of immunization. Tumor-bearing mice underwent three rounds of treatment with 100 μg of anti-PD-1 monoclonal antibody (BioLegend Cat# 11402, RRID:AB_313573), starting on day 20 post-implantation.

### 2.6. In Vivo Imaging System

Tumor burden was assessed before the PD-1 blockade therapy following the intraperitoneal injection of 200 μL luciferin (Regis Technologies, Morton Grove, IL, USA, Cat#1-360222-200), followed by imaging using IVIS Spectrum X5, Revvity with the Living Imaging 4.8.2 Software.

### 2.7. Skin, Lymph Node, and Tumor Processing

Skin and skin-draining lymph nodes (SDLNs) from immunized mice were collected at the time of sacrifice. SDLNs were macerated with a syringe piston and filtered through a 70 μm filter (Miltenyi Biotech, Bergisch Gladbach, Germany, Cat#130-095-823) using PBS. The Tumor Dissociation Kit from Miltenyi Biotec Inc., Bergisch Gladbach, Germany, (#130-095-929) was used to collect immunized skin to extract skin-infiltrating T cells for flow cytometry. Tumors were harvested and analyzed by IHC.

### 2.8. Ex Vivo ELISpot Assay

The ELISpot assay was performed using murine IFNγ capture and detection reagents by Mabtech, Cincinnati, OH, USA (Code: 3321-3PW), following the manufacturer’s instructions for a pre-coated ELISpot white plate. Splenocytes and lymphocytes (5 × 10^5^ cells) from SDLNs from citENO1- or unmodified ENO1 (umENO1)-vaccinated mice were cultured with citENO1 or umENO1 peptides (10 μg/mL) per pre-coated ELISpot well in quadruplicate. Splenocytes and lymphocytes from PBS-treated mice were used as additional controls. Isolated cells were subsequently pooled for each experimental arm (N = 3 mice per pool for cit-ENO1- or umENO1-vaccinated mice, and N = 4 mice for PBS-treated mice). The plates were incubated for 48 h at 37 °C in an atmosphere of 5% CO_2_. After incubation, the captured IFNγ was detected by biotinylated specific IFNγ antibodies and developed with streptavidin alkaline phosphatase and chromogenic substrate. Spots were analyzed and counted using the automated plate reader CTL Analyzer at the Oncology Research and Immuno-mONitoring core (ORION) at the MD Anderson Cancer Center. PMA (10 ng/mL) and ionomycin (1 μg/mL) were used as the positive controls. For MHC-blocking studies, 20 μg/mL of anti-CD8 (Bio X Cell Cat# BE0061, RRID:AB_1125541) and anti-CD4 (Bio X Cell Cat# BE0003-1, RRID:AB_1107636) antibodies were added to ELISpot assays.

### 2.9. Toxicology

B6129SF/J mice were sacrificed four weeks after the last round of immunization to collect organs and tissues for toxicity analysis. Brain, eye, intestine, spleen, stomach, heart, lung, kidney, liver, and knee biopsies were collected under an institutional review board protocol and archived as formalin-fixed, paraffin-embedded specimens. All samples were sent to the Research Histology Core Laboratory, MD Anderson Cancer Center, for the analysis of toxic-related morphological changes.

### 2.10. Flow Cytometry

Cells were stained with Ghost UV450 (Tonbo Cat#13-0868), CD45-Alexa 700 (BioLegend Cat#103128, RRID:AB_493715), CD8-APC-Cy (BioLegend Cat#100714, RRID:AB_312753), CD4-BV510 (BioLegend Cat#100553, RRID:AB_2562608), CD62L-BV650 (BioLegend Cat#104453, RRID:AB_2800559), CD44-PerCP-Cy5.5 (BioLegend Cat#103032, RRID:AB_2076206), PD-1-BV605 (BioLegend Cat#135219, RRID:AB_2562616), CD69-PE-Dazzle 594 (BioLegend Cat#104536, RRID:AB_2565583), CD25-PE-Cy7 (BioLegend Cat#101915, RRID:AB_2616762), and CD127- FITC (BioLegend Cat#121106, RRID:AB_493503). All the samples were run in a Fortessa X-20 cytometer (Beckton, Dickinson, Franklin Lakes, NJ, USA).

### 2.11. Statistics

Statistical analyses were performed using Prism 10.3.1 (GraphPad Software Inc., La Jolla, CA, USA). For flow cytometry, ELISpot, and H-score data, statistical significance was determined using reported Tukey multiple comparison testing and adjusted 2-sided *p*-values unless otherwise specified. For tumor growth studies, repeated measures two-way ANOVA was used to evaluate differences over time and between treatment groups. For survival analyses, statistical significance was determined using the Kaplan–Meier survival analysis and the Logrank (Mantel–Cox) test to compare two groups. Statistical significance was defined as * *p* ≤ 0.05, ** *p* ≤ 0.01, *** *p* ≤ 0.001, and **** *p* ≤ 0.0001.

## 3. Results

### 3.1. Citrullinated ENO1 Peptide-Based Vaccine

ENO1 is overexpressed in a wide range of tumors and its biological implications in cancer have been extensively studied [11,12,20,24,25,36,37]. We used mass spectrometry to identify citrullinated peptides from the surfaceome and immunopeptidome of breast cancer cell lines and in the immunoglobulin-bound (IgB) fraction from TNBC patient plasma (Figure 1A) for vaccine development. The significantly increased levels of ENO1 mRNA were found in tumor versus non-tumor tissues in the TCGA data (Figure 1B). ENO1 was confidently identified in BRCA wild-type and mutated cell lines (Figure 1C) with citrullination modifications (Figure 1D). Fourteen cit-peptides have been identified. The data analysis was performed according to the workflow presented in Table 1. Peptides between 22 to 24 amino acids in length including the MHC-II-binding immunocores were selected [38] and aligned with mouse ENO1 (P17182) to confirm high homology, and the same peptides searched against MHC-I were listed as well, to investigate the potential affinity. The peptides were synthesized and citrullinated in vitro. For vaccination, we designed Peptide 1 (sequence 32–56) identified from breast cancer cell surface analysis with 100% human to mouse homology, Peptide 2 (sequence 164–186) identified from plasma IgB fraction of TNBC patient plasmas has two amino acids 176A and 177N substituted with 176S in the mice [19], and Peptide 3 (sequence 262–284) identified from breast cancer cell surface proteome and mild acid elution peptidome encompassing one amino acid, 272 S, that substituted 272 T. The binding prediction of the vaccine peptides in the unmodified form searched using NetMHCIIpan and NetMHCpan is shown (Appendix A). Eight strong and one weak in Peptide 1, six strong and two weak in Peptide 2, and four strong in Peptide 3 were identified as MHC-II binders, whereas one strong and one weak in Peptide 1, three strong in Peptide 2, and two strong and two weak in Peptide 3 were identified as MHC-I binders. The RNA-seq gene expression of HLAs in breast cancer cell lines is shown in Appendix A.

### 3.2. PADI2 Is Correlated with Citrullination of ENO1 in Murine Breast Tumors

We analyzed the expression of PADI2, citENO1, and ENO1 in our TNBC model by IHC and found that all three targets were highly expressed in tumor tissues with respective H-scores of 215, 204, and 193 (Figure 2A and Appendix A). As a control, we analyzed the mammary fat pad of non-tumor-bearing mice, which showed the low expression of PADI2 and citENO1 and the moderate expression of ENO1 (Figure 2B and Appendix A). The further analysis of the H-score in different regions of the tumor and control tissues showed significant differences in PADI2 (*p* < 0.0001), citENO1 (*p* < 0.0001), and ENO1 (*p* = 0.0016) (Appendix A).

### 3.3. citENO1 Vaccine Impacts Tumor Growth and Survival

We next analyzed the acute immune response elicited by the citENO1 cancer vaccine and its potential toxicity. To this end, we immunized the mice thrice weekly with citrullinated or unmodified ENO1 peptides (Table 2) as the antigen (10 nmol), in combination with the TLR3 agonist Poly I:C as an adjuvant (10 μg), subcutaneously (s.c.) in female B6129SF1/J mice in the left flank (Figure 3A). Mice were sacrificed one week after the last immunization round, and their vital organs, as well as their SDLNs, and immunized skin were harvested. Using flow cytometry (Appendix A), we observed that citENO1 vaccination induced a larger percentage of activated CD8+ PD-1+ T cells in the SDLNs compared with the control groups and with unmodified ENO1, with *p* = 0.0128 between the adjuvant alone and citENO1 vaccine groups (Figure 3B, left), as well as for the activation marker CD69 (Appendix A, left). Remarkably, we observed similar results with CD8+ T effector memory (Tem), characterized based on CD44+ CD62L- markers, with *p* = 0.0473 between the citENO1 and umENO1 groups (Figure 3B, middle). We also observed the same trend when we compared the ratio of CD8 Tem vs. CD8 naïve, with a *p* = 0.056 between citENO1 and umENO1 groups (Figure 3B, right). CitENO1 peptide-1 immunization induced the highest percentage of CD8+ PD-1+ T cells in the immunized skin, *p* = 0.0126 between citENO1 vs. umENO1 groups (Figure 3C). Both the citENO1 and umENO1 vaccines induced a CD4+ PD-1+ T cell population in SDLNs (Figure 3D, left), a Tem response (Figure 3D, middle), and the ratio CD4 Tem vs. CD4 naïve in SDLNs (Figure 3D, right), and a CD4+ PD-1+ T cell population in the skin compared with the control groups PBS and Poly I:C (Figure 3E). Neither citrullinated nor unmodified ENO1 immunizations induced CD69 expression (Appendix A, right) or induced CD4+ Tregs compared with the PBS control (Appendix A), or central memory response characterized based on CD44+ CD62L+ (Tcm) in CD8 or CD4 T cells (Appendix A). Based on the inspection of the vital organs harvested, we observed that citENO1 and umENO1 vaccines did not show evidence of toxicity (Appendix A).

Then, we evaluated the immune response of the three umENO1 or citENO1 peptides combined, immunizing mice with the mixture of the three peptides with the same vaccine schedule (Figure 3A), and waiting for two weeks after the last round of immunization to sacrifice and harvest the SDLNs and skin. In the SDLNs, we observed an increase in PD1+ T cell among both CD8+ and CD4+ T cell populations following immunization with either umENO1 and citENO1 (Appendix A). Neither umENO1 nor citENO1 immunizations induced CD69 expression in T cells (Appendix A). Regarding the Tcm response in SDLNs, citENO1 immunization resulted in a higher percentage of Tcm among CD8+ T cells, though this difference was not statistically significant (*p* = 0.0860). No clear changes were observed in the CD4+ T cell central memory (Appendix A). We observed a reduced percentage of CD4+ Tregs in SDLNs in mice immunized with citENO1, compared with the control and umENO1 groups, although it was not significant (Appendix A). The analysis of PD1 expression in skin-infiltrating T cells revealed a higher percentage of PD1+ cells among CD8+ and CD4+ T cells (Appendix A). Both umENO1 and citENO1 immunizations led to an increased percentage of CD8+ Tcm, and citENO1 induced the highest percentage of CD4+ Tcm among the skin-infiltrating lymphocytes (Appendix A).

Next, we determined whether the immune response elicited by the citENO1 peptide vaccine could have an effective anti-tumor response. After four weeks of the last immunization round, we implanted 1.5 × 106 cells of the BRCA/TP53 syngeneic cell line in the mammary fat pad (Figure 3A). Only the citENO1 peptide vaccine significantly delayed tumor development, with a *p* value of <0.001 when compared to the citENO1 vaccine in all the other groups (Figure 3F), and improved overall survival (Figure 3G), with *p* = 0.0100 comparing citENO1 with umENO1 vaccines.

### 3.4. citENO1 Vaccination Induces citENO1-Specific IFNγ-Producing CD4+ T Cells

We further characterized the specificity of the response elicited by the citENO1 vaccine using an ex vivo ELISpot assay. For these studies, citENO1- or umENO1-immunized mice were sacrificed two weeks after the last round of vaccination, and isolated splenocytes and SDLN lymphocytes were cultured ex vivo with citENO1 peptides (Figure 4A) or umENO1 peptides (Figure 4B). The citENO1 peptide mix significantly increased IFNγ spot formation in splenocytes from citENO1-immunized mice compared to the media-only control (*p* = 0.0018). Splenocytes from umENO1-immunized mice also showed an increase in IFNγ spots following ex vivo culture with citENO1 peptides, albeit not statistically significant (*p* = 0.095) (Figure 4A). A similar response was observed when splenocytes were cultured with the umENO1 peptide mix (Figure 4B).

To determine which subset of T cells was responsible for IFNγ production, CD8 or CD4 blocking antibodies were added to the culture. IFNγ spots induced by the citENO1 mix from citENO1-immunized mice or umENO1-immunized mice splenocytes were significantly reduced when CD4 was blocked (*p* = 0.0014 and 0.0236, respectively) (Figure 4A); however, no reduction was observed when CD8 was blocked (*p* = 0.8464). A similar response was observed with the umENO1 peptide mix (Figure 4B).

### 3.5. citENO1 Vaccine Enhances Immunotherapy with PD-1 Blockade

IHC showed that PD-L1 was highly expressed in the BRCA/TP53 tumor tissue, showing a significant (*p*-value < 0.0001) difference in H-scores between PD-L1 and 2nd antibody only (Appendix A); thus, blocking the PD-1/PD-L1 axis may be an immunotherapy strategy for this TNBC model. Following our immunization protocol with citENO1 peptides in one arm and non-citrullinated ENO1 peptides in the other, we implanted BRCA/TP53-Luc one week after immunization and quantified tumors by IVIS on day 14 post implantation (Figure 5A and Appendix A). On the day of IVIS, both vaccines had significantly lower luminant intensity (photon/s) coming from the tumor than the PBS control group, with *p* < 0.0001 in both vaccines compared to the PBS control. CitENO1-immunized mice had a lower luminant intensity than umENO1-immunized, but the difference was not significant (Appendix A). Mice in each arm were divided into two groups with comparable tumor sizes for treatment with the PD-1 blockade or IgG control (Figure 5A). Five mice in the citENO1 arm were treated with the PD-1 blockade, which resulted in tumor eradication in four of five mice, with the slowing of tumor growth in the fifth mouse, and with a significant difference between umENO1 vaccine plus anti-PD1 and citENO1 vaccine plus anti-PD-1 (*p* = 0.0022) (Figure 5B,C). All five mice vaccinated with citENO1 and treated with anti-PD1 vaccine survived until sacrifice (Figure 5D). In contrast, tumors progressed in half of the mice in the citENO1 arm treated with the IgG control and were eradicated in the other half, and the tumor progressed in all mice immunized with the umENO1 vaccine (Figure 5D). The CitENO1 vaccine with and without the PD-1 blockade had significantly better survival than the umENO1 vaccine (*p* = 0.0460 and *p* = 0.0184, respectively) (Figure 5D).

## 4. Discussion

The antigenicity of citrullinated ENO1 protein and peptides is relevant in TNBC, given their occurrence as MHC-bound peptides and immunoglobulin-bound proteins in patient plasma [19]. The three synthesized citENO1 peptides, which were prioritized from our mass spectrometry data, plus Poly I:C as an adjuvant, elicited a robust CD8+ effector memory T cell response, activated CD8+ and CD4+ PD-1+ effector T cells in SDLNs and immunized skin, and a citENO1-specific IFNγ-producing CD4+ T cell response. This immune response elicited by the citENO1 vaccine delayed tumor growth compared to that elicited by unmodified ENO1 peptides. The citENO1 vaccine enhanced the anti-tumor response through the PD-1 blockade, mitigating PD-1+ T cells and PD-L1 expression in BRCA/TP53 tumors.

It has been observed that T cells from RA patients can identify umENO1 and citENO1 peptides with different affinities to HLA-DRB1*04:01, *05:04, and *01:01 [23]. Furthermore, some of these citENO1 peptides found in RA patients have previously been used as cancer vaccines [39]. It is worth mentioning that our selection of citENO1 peptides was based on evidence obtained from TNBC sources, such as MHC-bound peptides from breast cancer cell lines. Moreover, in a previous study, the adjuvant used was intended to induce a CD4+ T cell response [39]. In our study, the data obtained indicated that the citrullinated peptide vaccine induced both CD8+ and CD4+ T cells. In a previous study, a mouse model with MHC-I knockdown was used. More recently, the same research group demonstrated that homocitrullinated peptides could induce a CD8+ T cell response in a melanoma mouse model [40].

Among all adjuvants available for cancer vaccines, we selected the TL3 agonist Poly I:C because dendritic cell 1 (cDC1) expresses TLR3, and its activation results in the induction of cytotoxic CD8+ T cells, which are crucial for an anti-tumor response [41,42]. Notably, the activation of cDC1 also primes a CD4 response, which enhances the activity of cDC1 and CD8+ T cells [43]. Inducing both CD4+ and CD8+ responses would be expected to result in a more pronounced anti-cancer immune response compared to only a CD4+ or CD8+ T cell response. Thus, we used Poly I:C as a proof of concept to show that our citENO1 peptides could be CD8- and CD4-dependent. Moreover, a more stable version of this adjuvant, poly-ICLC, has already been used in a pilot study of breast cancer patients in a non-citrullinated peptide-based vaccine, showing modest immunogenicity, which may have been the result of dosing. Furthermore, TLR3 agonists have been shown to be safe and well tolerated in breast cancer settings [44]. However, comparing Poly I:C with other adjuvants and different dosages will give data to determine the best vaccine for future clinical trials. Since ENO1 is expressed on the tumor cell surface as a whole protein, inducing CD4+ T cells, the vaccine may also induce citENO1-specific B cells, resulting in antibody-dependent cellular cytotoxicity (ADCC) [45]. We previously found circulating citENO1-specific IgG in TNBC [19].

Our TNBC model has *BRCA1* genetically modified as a conditional knockout [35]; thus, it is plausible that our citENO1 vaccine may be personalized for *BRCA1*-mutated TNBC. *BRCA1* is a tumor suppressor gene linked to the inherited susceptibility to breast cancer [46]. Approximately 75–80% of *BRCA1* mutation-associated breast cancers are TNBC and basal-like [47,48]. Prenatal *BRCA1* epimutations also contribute to TNBC development, which may be the underlying cause of approximately 20% of TNBC and low-ER-expression breast cancers [49]. Interestingly, both *BRCA1* status and stromal TILs (sTILs) impact TNBC treatment response [50]. TNBC with low sTILs and a pathogenic germline *BRCA1* mutation have poor survival [50]. Blocking the PD-1/PD-L1 axis improves overall survival [51], which is affected by the response of TILs [52]. The citENO1 vaccine may enhance the efficacy of PD-1 inhibitor immunotherapy, which is currently FDA-approved for use in patients with TNBC [5,51,53].

Our findings suggests that our citENO1 vaccine could be considered for clinical application in high-risk women to prevent breast cancer formation, as well as for use in combination with immune checkpoint inhibitors (ICIs) for treatment. However, before it can be applied at the clinical level, several potential limitations must be considered. Our findings are based on studies in mice using peptides found in both humans and mice. Since the immune systems of humans and mice differ, the translational relevance of our results remains uncertain. For instance, we observed citENO1-specific T cells induced by our vaccine in genetically identical mice. Given the polymorphism of HLA class I and II molecules in the human population, this immune response may only occur in a subset of individuals. To begin addressing this issue, more human immunological data—such as peripheral blood mononuclear cells (PBMCs) or (TILs)—are needed to evaluate the capacity of human T cells to recognize citENO1 peptides from our vaccine, as well as their antigen-specific cytotoxic function and long-term memory potential. Furthermore, while our model is a *BRCA1*-mutant TNBC, the citENO1 vaccine could potentially be applicable to other breast cancer subtypes. Finally, although we observed no toxicity in mice, we currently lack long-term safety data and information on off-target autoimmunity in humans.

Our work suggests that the development of a cancer vaccine targeting novel post-translationally modified antigens in TNBC has the potential to enhance the efficacy of ICIs, considering the poor prognosis of TNBC [54].

## Figures and Tables

**Figure 1 vaccines-13-00629-f001:**
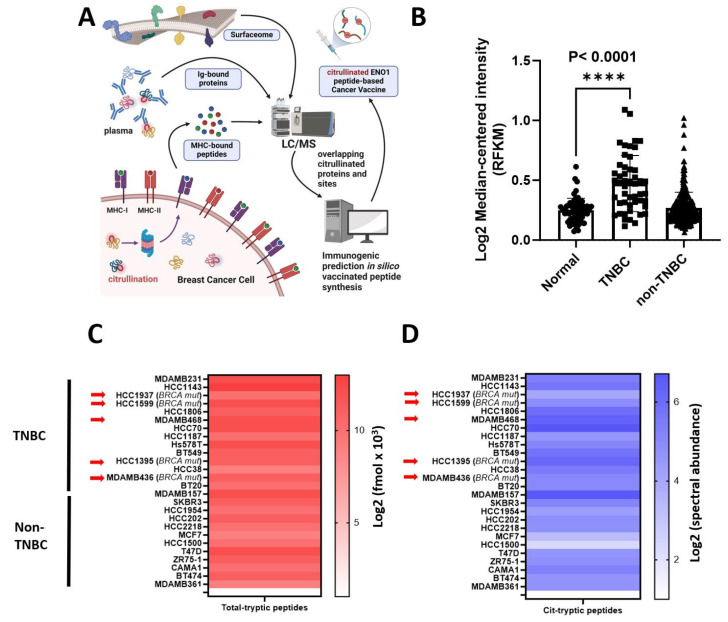
Cancer vaccine development and proteomic expression level of ENO1 and PADI2 in human breast cancer. (**A**) Proteomic strategy from different breast cancer sources (surface, IgG-bound protein from blood, and MHC-bound peptides from cell lines) to select citrullinated ENO1 peptides to develop a cancer vaccine. (**B**) Gene expression of ENO1 in TCGA. Log2 (normal-median intensity) = 0.2392, Log2 (TNBC-median intensity) = 0.4709, and Log2 (non-TNBC-median intensity) = 0.2390 (**C**) ENO1 fmol protein expression from total cell extract in breast cancer cell lines, and arrows indicate BRCA mutation. (**D**) Citrullinated ENO1 tryptic peptides identified from total cell extracts in breast cancer cell lines, and arrows indicate BRCA mutations. Statistical significance was defined as **** *p* ≤ 0.0001.

**Figure 2 vaccines-13-00629-f002:**
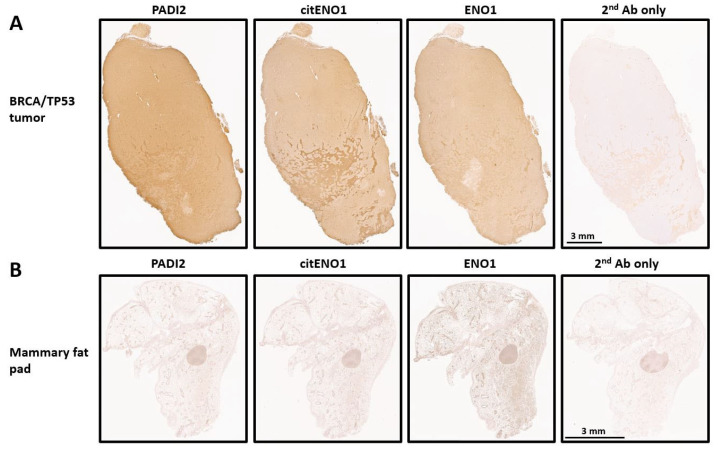
Citrullinated and unmodified ENO1 expression in breast tumor and normal tissue. IHC expression of PADI2, citENO1, ENO1, and 2nd antibody control in (**A**) breast cancer tissue section collected from BRCA/TP53-bearing mouse and (**B**) mammary fat pad tissue from mouse without tumor. Characteristic sample of tumor and fat mammary pad tissues are shown, and several sections for IHC were obtained from them.

**Figure 3 vaccines-13-00629-f003:**
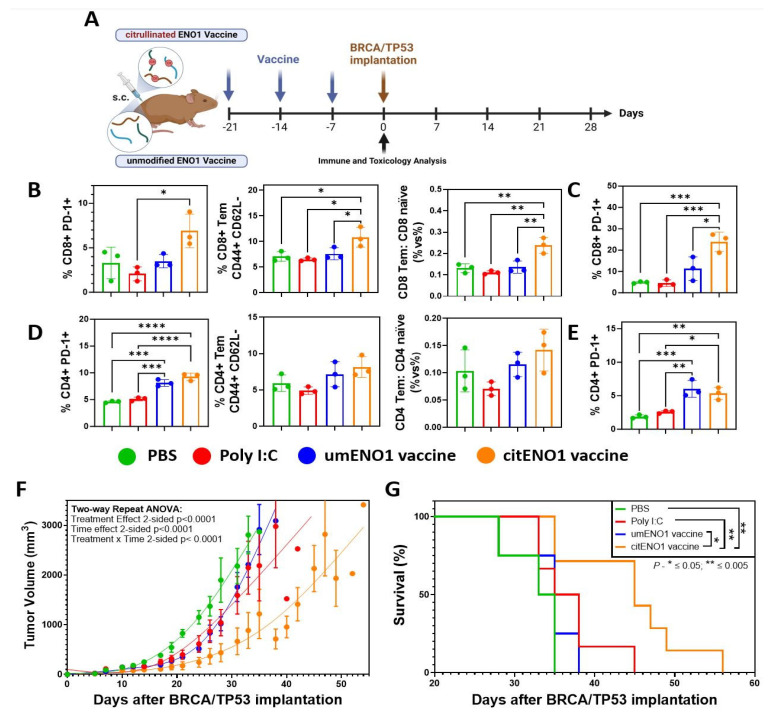
Citrullinated and unmodified ENO1 vaccine anti-tumoral immune response. (**A**) B6129SF1/J female mice were immunized s.c. three times with citrullinated or unmodified ENO1 peptide 1 plus Poly I:C, as well as control groups, with Poly I:C and PBS. After one week of the last immunization, the mice were sacrificed to harvest the draining lymph nodes and the immunized skin. (**B**) CD8+ PD-1+ T cell (left), CD8+ CD44+ CD62L− Tem cell (middle), and the CD8 Tem: CD8 naïve ratio (right) from draining lymph nodes of immunized mice, and (**C**) CD8+ PD-1+ T cell from immunized skin, analyzed by flow cytometry. (**D**) CD4+ PD-1+ T cell (left), CD4+ CD44+ CD62L− Tem cell (middle), and the CD4 Tem: CD4 naïve ratio (right) from draining lymph nodes of immunized mice, and (**E**) CD4+ PD-1+ T cell from immunized skin, analyzed by flow cytometry. Mice immunized with the same protocol as above but with peptides 1, 2, and 3 were implanted with *BRCA1*^co/co^; MMTV-Cre; p53^+/−^ cell line in the fat mammary pad four weeks after the last immunization, measuring the (**F**) tumor growth and (**G**) overall survival. Statistical significance was defined as * *p* ≤ 0.05, ** *p* ≤ 0.005, *** *p* ≤ 0.001, and **** *p* ≤ 0.0001.

**Figure 4 vaccines-13-00629-f004:**
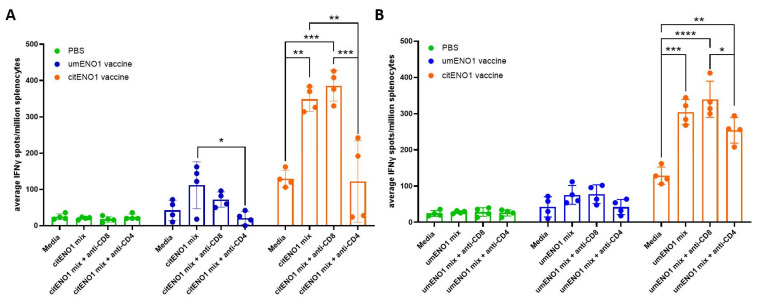
IFNγ responses to citrullinated and unmodified ENO1 peptides from immunized mice. Mice were immunized as in Figure 3A, and after two weeks, they were sacrificed and their spleen and SDLNs were harvested, and cultured with (**A**) citENO1 synthetic peptides, or (**B**) umENO1 synthetic peptides. Ex vivo responses to stimulation were assessed by IFNγ ELISpot, as well as the responses after the addition of anti-CD8 and anti-CD4 blocking antibodies. Statistical significance was defined as * *p* ≤ 0.05, ** *p* ≤ 0.005, *** *p* ≤ 0.001, and **** *p* ≤ 0.0001.

**Figure 5 vaccines-13-00629-f005:**
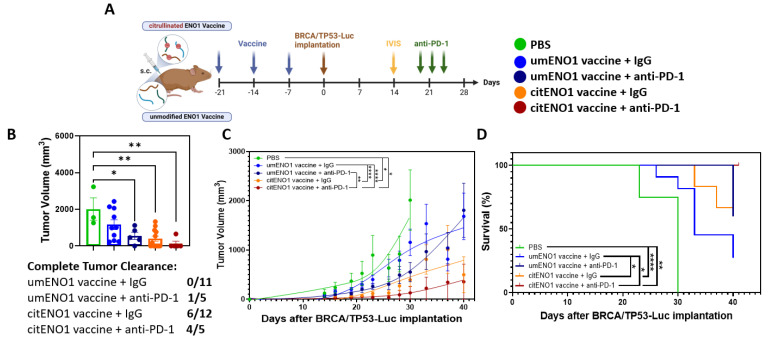
Citrullinated and unmodified ENO1 vaccine in combination with PD-1 blockade immunotherapy. (**A**) B6129SF1/J female mice were immunized s.c. three times with citrullinated or unmodified ENO1 vaccine, as well as the PBS control group, and after one week, they were implanted with BRCA/TP53-Luc-expressing cell line in the mammary fat pad. Two weeks after the tumor implantation, the tumor burden was assessed by IVIS to select the comparable groups for anti-PD-1 immunotherapy. At days 20, 23 and 26 after the implantation, the mice were injected with 100 ug of anti-PD-1 or IgG control antibody intratumorally. (**B**) The tumor volume was measured at day 30, and the complete response rate among the groups. The (**C**) tumor growth and (**D**) overall survival was measured until day 41, when mice were sacrificed. Statistical significance was defined as * *p* ≤ 0.05, ** *p* ≤ 0.005, and **** *p* ≤ 0.0001.

**Table 1 vaccines-13-00629-t001:** Discovery of citrullinated sites of ENO1 in TNBC and vaccination peptides design based on immunocore prediction.

Discovered Cit-Peptides and Conversion to Unmodified Sequence	Sequence Position	Cit-R Position	Discovery Source	Predicted Immunocore by NetMHCIIpan v3.2	Predicted Immunocore by NetMHCpan v4.1	Vaccine Peptides	Designed Vaccine Peptides	Designed Vaccine Peptides	Human to Mouse Homology	Designed Vaccine Peptides
ENO1-Human (P06733)							Human ENO1, unmodified form	Mouse Eno1, unmodified form		Mouse Eno1 cit-form
Step (i)				Step (ii)	Step (ii)		Step (iii)	Step (iv)		Step (v)
EIFDS(cit-R)GNPTVEVDLFTSK	10–28	15	Cell surface trypsin	DSRGNPTVE, RGNPTVEVD	SRGNPTVEV	Unused	Unused	Unused		Unused
FTSKGLF(cit-R)AAVPSGASTGIYE	25–45	32	Mild acid elution peptidome-no trypsin	RAAVPSGAS, AAVPSGAST, AVPSGASTG, VPSGASTGI, PSGASTGIY, GASTGIYEA, SGASTGIYE, STGIYEALE, TGIYEALEL, ASTGIYEAL, GIYEALELR	PSGASTGIY, VPSGASTGI, RAAVPSGAS	Vaccine peptide 1, 32–56	RAAVPSGASTGIYEALELRDNDKTR	RAAVPSGASTGIYEALELRDNDKTR	100%	(cit-R)AAVPSGASTGIYEALEL(cit-R)DNDKT(cit-R)
SKGLF(cit-R)AAVPSGASTGIYEAL	27–47	32	Mild acid elution peptidome-no trypsin	RAAVPSGAS, AAVPSGAST, AVPSGASTG, VPSGASTGI, PSGASTGIY, GASTGIYEA, SGASTGIYE, STGIYEALE, TGIYEALEL, ASTGIYEAL, GIYEALELR, RAAVPSGAS, AAVPSGAST, AVPSGASTG, VPSGASTGI, PSGASTGIY	PSGASTGIY, VPSGASTGI, RAAVPSGAS	Vaccine peptide 1, 32–56	RAAVPSGASTGIYEALELRDNDKTR	RAAVPSGASTGIYEALELRDNDKTR	100%	(cit-R)AAVPSGASTGIYEALEL(cit-R)DNDKT(cit-R)
AAVPSGASTGIYEALEL(cit-R)DNDK	33–54	50	Cell surface trypsin	GASTGIYEA, SGASTGIYE, STGIYEALE, TGIYEALEL, ASTGIYEAL, GIYEALELR	GIYEALELR, PSGASTGIY, VPSGASTGI	Vaccine peptide 1, 32–56	RAAVPSGASTGIYEALELRDNDKTR	RAAVPSGASTGIYEALELRDNDKTR	100%	(cit-R)AAVPSGASTGIYEALEL(cit-R)DNDKT(cit-R)
AVEKGVPLY(cit-R)HIADLAGNSE	123–142	132	Mild acid elution peptidome-no trypsin	LYRHIADLA	AVEKGVPLY	Unused	Unused	Unused		Unused
HIADLAGNSEVILPVPAFNVINGGSHAGNKLAMQEFMILPVGAANF(cit-R)EAMR	133–183	179	Plasma Ig-bound trypsin	AMQEFMILP, EFMILPVGA, FMILPVGAA, MILPVGAAN, ILPVGAANF, MQEFMILPV, QEFMILPVG, PVGAANFRE, VGAANFREA	FMILPVGAA, GAANFREAM, ILPVGAANF	Vaccine peptide 2, 164–186	AMQEFMILPVGAANFREAMRIGA	AMQEFMILPVGASSFREAMRIGA	176 A, 177 N (Human) to 176 S, 177 S (Mouse)	AMQEFMILPVGASSF(cit-R)EAM(cit-R)IGA
SGKYDLDFKSPDDPS(cit-R)YISPDQLADLYK	254–281	269	Cell surface trypsin	FKSPDDPSR, SRYISPDQL, RYISPDQLA, ISPDQLADL, LADLYKFKS	KSPDDPSRY, RYISPDQLA, SPDDPSRYI, SPDQLADLY, SRYISPDQL	Vaccine peptide 3, 262–284	KSPDDPSRYISPDQLADLYKSFV	KSPDDPSRYITPDQLADLYKSFV	272 S (Human) to 272 T (Mouse)	KSPDDPS(cit-R)YITPDQLADLYKSFV
SPDDPS(cit-R)YISPDQLADLYK	263–281	269	Cell surface trypsin	FKSPDDPSR, SRYISPDQL, RYISPDQLA, ISPDQLADL, LADLYKFKS	RYISPDQLA, SPDDPSRYI, SPDQLADLY, SRYISPDQL	Vaccine peptide 3, 262–284	KSPDDPSRYISPDQLADLYKSFV	KSPDDPSRYITPDQLADLYKSFV	272 S (Human) to 272 T (Mouse)	KSPDDPS(cit-R)YITPDQLADLYKSFV
FKSPDDPS(cit-R)YISPDQLADL	261–279	269	Mild acid elution peptidome-no trypsin	FKSPDDPSR, SRYISPDQL, RYISPDQLA, ISPDQLADL, LADLYKFKS	KSPDDPSRY, RYISPDQLA, SPDDPSRYI, SRYISPDQL	Vaccine peptide 3, 262–284	KSPDDPSRYISPDQLADLYKSFV	KSPDDPSRYITPDQLADLYKSFV	272 S (Human) to 272 T (Mouse)	KSPDDPS(cit-R)YITPDQLADLYKSFV
KSPDDPS(cit-R)YISPDQLADL	262–279	269	Mild acid elution peptidome-no trypsin	FKSPDDPSR, SRYISPDQL, RYISPDQLA, ISPDQLADL, LADLYKFKS	RYISPDQLA, SPDDPSRYI, SRYISPDQL	Vaccine peptide 3, 262–284	KSPDDPSRYISPDQLADLYKSFV	KSPDDPSRYITPDQLADLYKSFV	272 S (Human) to 272 T (Mouse)	KSPDDPS(cit-R)YITPDQLADLYKSFV
LAQANGWGVMVSH(cit-R)SGETEDTF	359–380	372	Mild acid elution peptidome-no trypsin	ANGWGVMVS, NGWGVMVSH, WGVMVSHRS, GVMVSHRSG	AQANGWGVM	Unused	Unused	Unused		Unused
SGETEDTFIADLVVGLCTGQIKTGAPCRSE(cit-R)	373–403	403	Cell surface trypsin	TEDTFIADL, TFIADLVVG, FIADLVVGL, IADLVVGLC, VVGLCTGQI, CTGQIKTGA	FIADLVVGL, TEDTFIADL, GETEDTFIA	Unused	Unused	Unused		Unused
YNQLL(cit-R)IEEELGSK	407–420	412	Cell surface trypsin	No hit	QLLRIEEEL	Unused	Unused	Unused		Unused
IEEELGSKAKFAGRNF(cit-R)NPLAK (C-terminal)	413–434	429	Mild acid elution peptidome-no trypsin	AKFAGRNFR, GRNFRNPLA, GSKAKFAGR, AGRNFRNPL	AGRNFRNPL, EELGSKAKF, GRNFRNPLA, IEEELGSKA, KAKFAGRNF, RNFRNPLAK	Unused	Unused	Unused		Unused

**Table 2 vaccines-13-00629-t002:** ENO1 mouse vaccination peptides in unmodified and citrullinated forms.

	Position	Cit-Sites	Length	Synthesized Vaccination Peptide Sequence
Peptide 1_wt	32–56	32, 50, 56	24	RAAVPSGASTGIYEALELRDNDKTR
Peptide 1_cit	32–56	32, 50, 56	24	(cit-R)AAVPSGASTGIYEALEL(cit-R)DNDKT(cit-R)
Peptide 2_wt	164–186	179, 183	22	AMQEFMILPVGASSFREAMRIGA
Peptide 2_cit	32–56	179, 183	22	AMQEFMILPVGASSF(cit-R)EAM(cit-R)IGA
Peptide 3_wt	262–284	269	22	KSPDDPSRYITPDQLADLYKSFV
Peptide 3_cit	32–56	269	22	KSPDDPS(cit-R)YITPDQLADLYKSFV

## Data Availability

The original contributions presented in this study are included in the article/Appendix A. Further inquiries can be directed to the corresponding author.

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
