# Peer review of "Citrullinated ENO1 Vaccine Enhances PD-1 Blockade in Mice Implanted with Murine Triple-Negative Breast Cancer Cells"

_vaccines, 2025, doi:10.3390/vaccines13060629_

Round 1

Reviewer 1 Report

Comments and Suggestions for Authors

This manuscript addresses a timely and highly relevant topic by investigating a novel cancer vaccine targeting post-translationally modified citrullinated ENO1 peptides in a triple-negative breast cancer (TNBC) mouse model. The study shows that immunization with citrullinated ENO1 (citENO1) peptides elicits robust T-cell responses, delays tumor growth, and enhances response to PD-1 immune checkpoint inhibition.

The experimental design is overall solid, the animal model is appropriate, and the integration with PD-1 blockade provides translational relevance. However, several major issues need to be addressed before this manuscript can be considered for publication.

Major Comments:

  1. Translational Relevance to Human TNBC Patients

While the preclinical findings are promising, the translational bridge to human disease is insufficiently developed. The manuscript lacks data (or discussion) regarding human T-cell recognition of citENO1 peptides, their HLA presentation potential, or patient-derived immune responses.

Revision Requested: Please provide supporting data (if available) or a discussion on whether citENO1 peptides are recognized by human T cells in TNBC patients. Alternatively, highlight this as a limitation.

  1. Peptide Selection Criteria Need Clarification

The selection of three citrullinated peptides from a pool of 14 is not well justified. The rationale based on MHC binding prediction, abundance, immunogenicity scores, or surface presentation should be clearly presented.

Revision Requested: Add a table or figure detailing the MHC-I/II binding predictions (IC50, %Rank), expression levels, and surface localization evidence for all candidate peptides and explain why these three were prioritized.

  1. Control Groups and Functional Validation

While the study includes controls with unmodified peptides and adjuvant alone, more functional validation of immune responses in control arms is warranted. The umENO1-induced responses are partially characterized but not clearly differentiated.

Revision Requested: Provide a deeper analysis of CD4/CD8 T cell subsets, cytokine expression, and memory phenotype in the umENO1 group. Consider including granzyme B or cytotoxicity assays if available.

  1. Absence of Direct Cytotoxicity Data

The manuscript infers cytotoxic CD8+ T cell activation from PD-1 and memory markers. However, direct evidence of cytolytic activity (e.g., in vitro killing assays, granzyme B, or perforin expression) is missing.

Revision Requested: If available, please include data to demonstrate antigen-specific cytotoxic function. Otherwise, acknowledge this limitation in the discussion.

  1. Clinical Applicability and Human Adaptation

Although the peptides show high murine/human homology, the clinical adaptability is not addressed in terms of manufacturability, safety, or regulatory path.

Revision Requested: Include a discussion on how these findings may translate to human clinical application. Mention challenges such as HLA variability, peptide synthesis, and adjuvant selection.

  1. In line 117, the inclusion of the PMID number (PMID: 31114875) appears to be redundant, as the sentence already cites reference [35] for the GEPIA2 web server. The authors should clarify the rationale for including the PMID here, especially since proper citation through reference numbering is already provided.

  1. The Methods - Proteomic analysis section lacks a clear explanation of how and why the peptides were selected. The authors should elaborate on the selection criteria, including MHC-I/II binding scores, surface presentation, expression levels, or any other relevant parameters used in the selection process.

  1. It would be helpful to include the country of origin (manufacturer's country) for the instruments and products used in the study to enhance reproducibility and transparency.

  1. In the Results section, the placement of Figure 2 (line 306) seems to interrupt the flow of the paragraph. It would be more appropriate to position the figure legend above the figure itself, rather than embedding it within the paragraph, to improve readability and overall clarity.

  1. In some of the analyses, p-values are reported; however, the specific statistical tests used are not clearly stated. The Methods section should clearly describe the statistical analyses performed, including the type of tests used, assumptions regarding normality, methods for group comparisons, and, if applicable, any variance analyses or corrections for multiple comparisons.

  1. While the synthesis of the peptides is described as customized from CPC Scientific, the manuscript lacks critical information regarding peptide purity (e.g., percentage) and quality control parameters, such as analytical HPLC and mass spectrometry (MS) data. Inclusion of these details is essential to ensure the reproducibility and reliability of the experimental results.

Minor Comments:

  1. Ensure that all statistical methods are explicitly described (some flow cytometry analyses refer to p-values but not always the test used).
  2. Figures are generally clear, but labeling could be improved for better readability, especially in complex flow plots.
  3. Expand abbreviations on first use (e.g., Tem, SDLN) to ensure clarity for general readers.

Author Response

Major Comments:

Comment 1: Translational Relevance to Human TNBC Patients

While the preclinical findings are promising, the translational bridge to human disease is insufficiently developed. The manuscript lacks data (or discussion) regarding human T-cell recognition of citENO1 peptides, their HLA presentation potential, or patient-derived immune responses.

Revision Requested: Please provide supporting data (if available) or a discussion on whether citENO1 peptides are recognized by human T cells in TNBC patients. Alternatively, highlight this as a limitation.

Response 1: Thank you bringing this issue up, and we have added further information about it in the Discussion, lines 480-487.

Comment 2: Peptide Selection Criteria Need Clarification

The selection of three citrullinated peptides from a pool of 14 is not well justified. The rationale based on MHC binding prediction, abundance, immunogenicity scores, or surface presentation should be clearly presented.

Revision Requested: Add a table or figure detailing the MHC-I/II binding predictions (IC50, %Rank), expression levels, and surface localization evidence for all candidate peptides and explain why these three were prioritized.

Response 2: Thank you for your suggestion. The peptides of origin were described “Discovery source” column (Table 1) which explains “Cell surface- trypsin”, “Mild acid elution peptidome-no trypsin” and “Plasma IgBound-trypsin” identifications.

Comment 3: Control Groups and Functional Validation

While the study includes controls with unmodified peptides and adjuvant alone, more functional validation of immune responses in control arms is warranted. The umENO1-induced responses are partially characterized but not clearly differentiated.

Revision Requested: Provide a deeper analysis of CD4/CD8 T cell subsets, cytokine expression, and memory phenotype in the umENO1 group. Consider including granzyme B or cytotoxicity assays if available.

Response 3: Regarding this point we have added the analysis of central memory response in the umENO1 and citENO1 vaccine (Supplementary Figure 4C), and also the CD4/CD8 analysis of activation and memory using the mix of peptides after two weeks of the last round of immunization (Supplementary Figure 5). We add the corresponding mentioning in the Results section.

Comment 4: Absence of Direct Cytotoxicity Data

The manuscript infers cytotoxic CD8+ T cell activation from PD-1 and memory markers. However, direct evidence of cytolytic activity (e.g., in vitro killing assays, granzyme B, or perforin expression) is missing.

Revision Requested: If available, please include data to demonstrate antigen-specific cytotoxic function. Otherwise, acknowledge this limitation in the discussion.

Response 4: This is a really relevant point, and because of that we are planning to do it for a follow-up article about citENO1, so we acknowledge this limitation in the discussion as asked in lines 483-487.

Comment 5: Clinical Applicability and Human Adaptation

Although the peptides show high murine/human homology, the clinical adaptability is not addressed in terms of manufacturability, safety, or regulatory path.

Revision Requested: Include a discussion on how these findings may translate to human clinical application. Mention challenges such as HLA variability, peptide synthesis, and adjuvant selection.

Response 5: Thank you for bringing this up, since our ultimate goal is to bring our vaccine to the clinic. We include these points in the discussion, lines 508-512.

Comment 6: In line 117, the inclusion of the PMID number (PMID: 31114875) appears to be redundant, as the sentence already cites reference [35] for the GEPIA2 web server. The authors should clarify the rationale for including the PMID here, especially since proper citation through reference numbering is already provided.

Response 6: We agree, it is redundant, we just provided the reference using the number, 35.

Comment 7: The Methods - Proteomic analysis section lacks a clear explanation of how and why the peptides were selected. The authors should elaborate on the selection criteria, including MHC-I/II binding scores, surface presentation, expression levels, or any other relevant parameters used in the selection process.

Response 7: Thank you for your suggestion. We have added the descriptions in the Method to explain the procedure, confidence of the identification and the peptide selection criteria, now in 2.1 Proteomic analysis section. (lines 91-128)

Comment 8: It would be helpful to include the country of origin (manufacturer's country) for the instruments and products used in the study to enhance reproducibility and transparency.

Response 8: We added the cytometer and its manufacturer were all the samples were run, as well as the country of origin of the MS equipment. All the reagents had their RRID:AB code, for transparency and easy finding.

Comment 9: In the Results section, the placement of Figure 2 (line 306) seems to interrupt the flow of the paragraph. It would be more appropriate to position the figure legend above the figure itself, rather than embedding it within the paragraph, to improve readability and overall clarity.

Response 9: Thank you for pointing this out. now the Figure Legend goes right after the Figure, just like the other ones.

Comment 10: In some of the analyses, p-values are reported; however, the specific statistical tests used are not clearly stated. The Methods section should clearly describe the statistical analyses performed, including the type of tests used, assumptions regarding normality, methods for group comparisons, and, if applicable, any variance analyses or corrections for multiple comparisons.

Response 10: To make the statistics used clearer in the Methods, we expand our explanation in this section, which is the last of the last, 2.11 Statistics. (lines 258-265)

Comment 11: While the synthesis of the peptides is described as customized from CPC Scientific, the manuscript lacks critical information regarding peptide purity (e.g., percentage) and quality control parameters, such as analytical HPLC and mass spectrometry (MS) data. Inclusion of these details is essential to ensure the reproducibility and reliability of the experimental results.

Response 11: We add the Certificate of Analysis, which has HPLC and MS data for each peptide, as the Supplementary Figure 1.

Minor Comments: 

Comment 12: Ensure that all statistical methods are explicitly described (some flow cytometry analyses refer to p-values but not always the test used).

Response 12: The test used now is found in the Methodology section, under 2.11 Statistics, so we avoid repeating that during the Results.

Comment 13: Figures are generally clear, but labeling could be improved for better readability, especially in complex flow plots.

Response 13: Thank you for mentioning this, and we have improved the Figures, especially Supplementary Figure 3 when we show the flow plots and all the gathering, labelling more appropriately.

Comment 14: Expand abbreviations on first use (e.g., Tem, SDLN) to ensure clarity for general readers.

Response 14: We make sure that all abbreviations are explained on the first use, and if the first use it was on the Abstract it was explained again in the main text.

Reviewer 2 Report

Comments and Suggestions for Authors

In the manuscript "A citrullinated-ENO1 vaccine enhances PD-1 blockade in mice implanted with murine triple-negative breast cancer cells”, León-Letelier et al. showed that citENO1 vaccine delayed tumor growth and prolonged overall survival, and therefore, the development of a cancer vaccine targeting novel post-translational modified antigens in TNBC has the potential to enhance the efficacy of immune checkpoint inhibitor. This manuscript is of scientific and practical interest, and well structured and written. To the opinion of this reviewer, this manuscript can be acceptable as it is for publication in Vaccines. To enhance the readership, the following revisions may be considered. Similar conclusion has been reached before (Refs 11 and 12). Given that these works already showed the immunogenicity of citrullinated ENO1, the authors should provide clearly what is novel in this study. In addition, the long-term memory induced by vaccination is worth of determining for further application. Refs citations should follow mdpi style.

Author Response

Comment 1: In the manuscript "A citrullinated-ENO1 vaccine enhances PD-1 blockade in mice implanted with murine triple-negative breast cancer cells”, León-Letelier et al. showed that citENO1 vaccine delayed tumor growth and prolonged overall survival, and therefore, the development of a cancer vaccine targeting novel post-translational modified antigens in TNBC has the potential to enhance the efficacy of immune checkpoint inhibitor. This manuscript is of scientific and practical interest, and well structured and written. To the opinion of this reviewer, this manuscript can be acceptable as it is for publication in Vaccines. To enhance the readership, the following revisions may be considered. Similar conclusion has been reached before (Refs 11 and 12). Given that these works already showed the immunogenicity of citrullinated ENO1, the authors should provide clearly what is novel in this study. In addition, the long-term memory induced by vaccination is worth of determining for further application. Refs citations should follow mdpi style

Response 1: Thank you for suggesting this, and to address it we have expanded in the second paragraph of the Discussion about the novelty on our citENO1 peptides compared with previous work. Also, we mentioned in the discussion that is important for further experiments determine the long-term memory. We have downloaded the MDPI style and use it for the referencing, thank you for point this out.

Round 2

Reviewer 1 Report

Comments and Suggestions for Authors

Major Comment – Lack of Discussion of Study Limitations

While the manuscript presents compelling preclinical data supporting the immunogenicity and therapeutic potential of a citrullinated ENO1 peptide-based vaccine in a murine triple-negative breast cancer (TNBC) model, an important omission is the lack of a discussion on study limitations. Addressing potential limitations is essential for a transparent and balanced scientific presentation, especially in translational cancer immunotherapy research.

Specifically, I recommend that the authors incorporate a brief paragraph near the end of the Discussion section to acknowledge and reflect upon the following limitations:

  1. Species Translation: The current findings are based solely on murine models. While informative, mouse immune systems differ significantly from humans, and translational relevance remains uncertain.
  2. HLA Variability: The immunogenicity and MHC binding of citENO1 peptides may vary considerably across human populations due to HLA class II polymorphism, which could impact vaccine efficacy in clinical settings.
  3. Lack of Human Immune Data: The study does not include validation of T cell responses using human peripheral blood mononuclear cells (PBMCs) or tumor-infiltrating lymphocytes (TILs). Demonstrating that human T cells can recognize these peptides would significantly strengthen the translational potential.
  4. Subtype Specificity: The tumor model used is BRCA1-mutant TNBC, and generalizability to other TNBC subtypes or hormone receptor-positive tumors is not established.
  5. Autoimmunity Consideration: Although acute toxicity was not observed in mice, long-term safety and the risk of off-target autoimmunity due to citrullinated peptide recognition remain unaddressed.

Including these points would not diminish the value of the current work but would enhance its credibility and help readers interpret the findings within an appropriate translational context.

Author Response

Comment 1: While the manuscript presents compelling preclinical data supporting the immunogenicity and therapeutic potential of a citrullinated ENO1 peptide-based vaccine in a murine triple-negative breast cancer (TNBC) model, an important omission is the lack of a discussion on study limitations. Addressing potential limitations is essential for a transparent and balanced scientific presentation, especially in translational cancer immunotherapy research.

Specifically, I recommend that the authors incorporate a brief paragraph near the end of the Discussion section to acknowledge and reflect upon the following limitations:

  1. Species Translation: The current findings are based solely on murine models. While informative, mouse immune systems differ significantly from humans, and translational relevance remains uncertain.
  2. HLA Variability: The immunogenicity and MHC binding of citENO1 peptides may vary considerably across human populations due to HLA class II polymorphism, which could impact vaccine efficacy in clinical settings.
  3. Lack of Human Immune Data: The study does not include validation of T cell responses using human peripheral blood mononuclear cells (PBMCs) or tumor-infiltrating lymphocytes (TILs). Demonstrating that human T cells can recognize these peptides would significantly strengthen the translational potential.
  4. Subtype Specificity: The tumor model used is BRCA1-mutant TNBC, and generalizability to other TNBC subtypes or hormone receptor-positive tumors is not established.
  5. Autoimmunity Consideration: Although acute toxicity was not observed in mice, long-term safety and the risk of off-target autoimmunity due to citrullinated peptide recognition remain unaddressed.

Including these points would not diminish the value of the current work but would enhance its credibility and help readers interpret the findings within an appropriate translational context.

Response 1: Thank you for bringing this up. Although we mentioned some of these issues in the second paragraph, we agree that your idea is better. So, we make a paragraph just about our limitations as asked for. We modified some of the other part of the discussion to give it more clarity and flow. All the changes are highlighted.